# Accurate estimation of cell-type composition from gene expression data

Daphne Tsoucas[1,2], Rui Dong[1,2], Haide Chen[3], Qian Zhu[1,2], Guoji Guo[3] & Guo-Cheng Yuan [ID] [1,2]

The rapid development of single-cell transcriptomic technologies has helped uncover the cellular heterogeneity within cell populations. However, bulk RNA-seq continues to be the main workhorse for quantifying gene expression levels due to technical simplicity and low cost. To most effectively extract information from bulk data given the new knowledge gained from single-cell methods, we have developed a novel algorithm to estimate the cell-type composition of bulk data from a single-cell RNA-seq-derived cell-type signature. Comparison with existing methods using various real RNA-seq data sets indicates that our new approach is more accurate and comprehensive than previous methods, especially for the estimation of rare cell types. More importantly, our method can detect cell-type composition changes in response to external perturbations, thereby providing a valuable, cost-effective method for dissecting the cell-type-specific effects of drug treatments or condition changes. As such, our method is applicable to a wide range of biological and clinical investigations.

[1] Department of Biostatistics and Computational Biology, Dana-Farber Cancer Institute, Boston, MA 02115, USA. [2] Department of Biostatistics, Harvard T.H. Chan School of Public Health, Boston, MA 02115, USA. [3] Center for Stem Cell and Regenerative Medicine, Zhejiang University School of Medicine, Hangzhou, China. Correspondence and requests for materials should be addressed to D.T. (email: dtsoucas@gmail.com) or to G.-C.Y. (email: gcyuan@jimmy.harvard.edu)

Gene expression profiling is widely used in biology and medicine for the systematic characterization of cellular or disease states. Identifying gene expression changes across conditions can help generate hypotheses as to underlying biological mechanisms. However, one common problem is that each sample has considerable cellular heterogeneity that bulk RNA-seq methods are not able to capture. As the overall signature generated from these methods only measures the average behavior, it is often the case that changes in gene expression only reflect changes in cell-type composition, rather than fundamental changes in cell states[1]. To alleviate such problems, a series of computational methods have been developed with the common goal of estimating the cell-type composition within a tissue sample from bulk RNA-seq data[2,3]. These methods, often referred to as deconvolution methods, provide an important means to distinguishing between changes in cell-type composition and changes in cell-state. Various estimation approaches have been used, including least squares regression[4], constrained least squares regression[5], quadratic programming[6–8], and ν-support vector regression[9].

However, existing methods have a number of important limitations. Most importantly, the underlying cell-type signatures must be known in advance. Most studies assume that such signatures can be identified from the bulk transcriptomic profiling of purified cell types. The success of cell-type purification relies heavily on the knowledge of specific markers as well as the ability to isolate cells from surrounding tissues. Moreover, it is now known that even the 'purified' cells may still contain significant cellular heterogeneity[10].

Recent single-cell transcriptomic methods[11,12] have provided a powerful approach to systematically characterizing cellular heterogeneity, thereby enabling the identification of new cell types/states and the reconstruction of developmental trajectories. Applications of single-cell methods in medicine have led to novel insights into disease progression and drug response[13–15]. Single-cell data provide an alternative approach to deriving cell-type signatures. In fact, a few recent studies[16,17] have extended deconvolution methods by estimating cell-type signatures from single-cell data, where cell types are inferred by clustering. While these methods are useful, a number of significant challenges remain. In particular, their estimates tend to be biased against cell types that either (1) make up a small proportion of the total bulk cell population, or (2) are characterized by lowly expressed genes. To remove these biases, we develop a cell-type-sensitive method for the estimation of the underlying cell fractions, using a novel weighted least squares approach.

## Results

**A weighted least squares approach to deconvolution.** We aimed to build a method that can accurately and comprehensively estimate the relative abundance of both common and rare cell types within a bulk sample. Much like recent studies[16,17], we use single-cell RNA-seq data to extract cell-type-specific gene expression signatures. Simply, the cell types are identified by clustering analysis. For each cell type, marker genes are identified by differential expression analysis, after which gene expression levels for each of these genes are averaged across all cells associated with the cell type. This results in a gene by cell-type signature matrix, which is denoted by **S** (see Methods section for details).

In order to accurately and comprehensively estimate the cell-type composition, we made a number of significant modifications to the standard ordinary least squares (OLS) approach, which underlies most existing methods[4–8]. In this approach, the deconvolution problem is represented as a system of linear equations: $\mathbf{Sx} = \mathbf{t}$, where $\mathbf{S}$ is an $n \times k$ gene signature matrix ($n =$ number of genes, $k =$ number of cell types), $\mathbf{t}$ is an $n \times 1$ vector representing the bulk RNA-seq data, and $\mathbf{x}$ is a $k \times 1$ vector containing the cell-type numbers. Since typically $n \gg k$, this is an over-determined equation with no exact solution. In the OLS approach, the solution $\mathbf{x}$ minimizes the total squared absolute error. This leads to two undesirable consequences. First, the estimation error for rare cell types is typically large since such a term has little impact on the total estimation error. Second, not all informative genes are effectively taken into account. The contribution of a gene can be minimal if its mean expression level is low, even if it is highly differentially expressed between different cell types.

To illustrate these effects, we carried out a highly idealized simulation. We generated a single-cell data set consisting of three cell types, each characterized by two differentially expressed marker genes. A portion of the data was used to create the signature matrix, while a non-overlapping portion was used to create the bulk data by averaging gene expression values across the cells. First, to see how the OLS formulation affects rare cell-type estimation, we varied the abundance of one cell type from 0.02% to 33.3% (see Methods section for details). When the abundance is very low, the relative percent error (RPE) of estimation, defined as $\mathrm{RPE} = \dfrac{\left| \frac{x_l}{\sum_{j=1}^{k} x_j} - \frac{\hat{x}_l}{\sum_{j=1}^{k} \hat{x}_j} \right|}{\frac{x_l}{\sum_{j=1}^{k} x_j}} * 100$ for cell type $l$ is very high (Fig. 1a), supporting our intuition that the OLS framework is not appropriate for estimating the prevalence of rare cell types. In addition, we varied the mean gene expression level of the two highly differentially expressed genes (fold change = 10) pertaining to one cell type such that the ratio of mean expression level between genes in this cell type vs. the other two cell types ranges from 0.001 to 0.2. As expected, the deconvolution accuracy is significantly affected by the mean expression level of these genes (Fig. 1b).

To mitigate these issues, we designed a weighted least squares approach to properly adjust the contribution of each gene. Accordingly, the weighted error term becomes: $\mathrm{Err} = \sum_{i=1}^{n} w_i \left( t_i - (\mathbf{Sx})_i \right)^2$. Our mathematical derivation indicates that setting $w_i = \frac{1}{(\mathbf{Sx})_i^2}$ optimally reduces the biases (see Methods section for details). To test this idea empirically, we applied this weighted approach to analyze the aforementioned simulated data. It is clear that both biases are significantly reduced (Fig. 1). Of note, we make the commonly used simplifying assumption that the total amount of RNA is approximately equal in each cell. If this is not true, the estimated contribution of each cell type may deviate from the actual cell abundance.

When applying our weighted least squares method in all real applications, we make a few adjustments required to make the weighting formulation tractable in all situations. Given that the weights are a function of the solution, we use an iterative method in which weights are initialized according to the solution from the unweighted method, then subsequently updated by the weighted least squares solution until convergence (see Methods section for details). Of note, while there is no theoretical guarantee that the converged solution reaches the global minimum, we find that in practice different initializations often end up at the same result, as demonstrated by our analysis of an intestinal stem-cell (ISC) data set described later (Supplementary Fig. 1). Next, given that cell-type proportions must be non-negative, the weighted least squares solution is constrained such that $x_j \geq 0$, for all $j$ cell types. Finally, a dampening constant is introduced to prevent

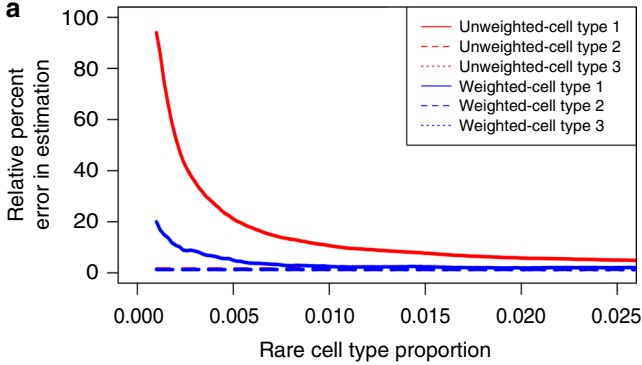

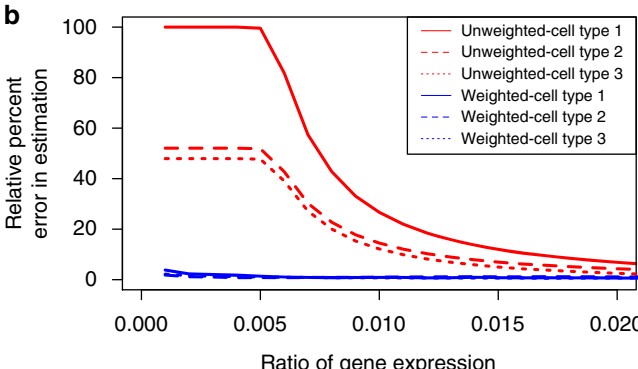

**Fig. 1** A simple simulation shows the advantages of a weighted least squares method. **a** A plot of relative percent error in estimation using both unweighted and weighted least squares approaches, for each of three cell types across various proportions of cell type 1, the rare cell type. Because of the increased influence of rare-cell-type-specific marker genes in the weighted sum of squares error, the weighted least squares method performs better in the estimation of rare cell types than the unweighted method. **b** A plot of relative percent error in estimation using both unweighted and weighted least squares approaches, for each of three cell types across various ratios of mean gene expression level between marker genes of cell type 1 and marker genes of cell types 2 and 3. Because of the increased influence of lowly expressed marker genes in the weighted sum of squares error, the weighted least squares method performs better in the estimation of all cell types than the unweighted method

infinite weights resulting from low cell-type proportions and/or low marker gene expression, which will lead to unstable solutions driven by only one or a few genes (see Methods section for details). Because of this last step, we subsequently refer to our method as dampened weighted least squares (DWLS).

**Benchmarking of DWLS on simulated PBMC data.** To evaluate the performance of our DWLS method, we first considered a benchmark data set introduced by Schelker et al.[17], who were among the first to consider the application of a single-cell derived gene expression signature to the problem of deconvolution. This data set is a compilation of 27 single-cell data sets from immune and cancer cell populations, derived from human donor peripheral blood mononuclear cells (PBMCs), tumor-derived melanoma patient samples, and ovarian cancer ascites samples. Since no bulk data was provided, we created 27 simulated bulk data sets by averaging expression values for each gene across all cells obtained from each donor, assuming that the bulk data is equivalent to the pooled data from individual cells. A similar assumption was made previously[17]. In addition, the cell-type-specific gene expression matrix was estimated by clustering the combined 27 single-cell

data sets. Marker genes were then chosen to match the genes used in the immune-cell-specific signature from CIBERSORT[9], and expression values for each marker gene were averaged within each cell type.

We applied ν-support vector regression (ν-SVR), quadratic programming (QP) and DWLS to the deconvolution of these 27 simulated bulk data sets. To quantify the overall performance of each method, we use two metrics. The first is a modified relative percent error metric, which quantifies the difference in true and estimated cell-type proportions, normalized by the mean of true and estimated cell-type proportions (see Methods section for details). Averaged across all cell types, the modified relative percent error is lowest for DWLS, at 53.3%, second lowest for ν-SVR, at 57.0%, and highest for QP, at 62.9%. The second is a more standard metric of absolute error between estimated and true cell-type proportions, in which we can see that absolute errors across cell types are again on average lowest for DWLS (Supplementary Table 1).

We further compared the accuracy of different methods on a per-cell-type basis (Fig. 2a). While ν-SVR performs well for the largest cell subpopulation, DWLS performs better over a wide range of cell types, especially the rarest cell groups. In particular, DWLS preserves a good balance between rare and common cell-type estimation. A similar trend can be seen from the standpoint of absolute error (Supplementary Table 1).

We took a closer look at the two rarest cell types across the 27 samples: dendritic and endothelial cells. Dendritic cells contribute to a maximum of 4.89% of the total cells in any given sample, with an average 0.999% prevalence across samples. Endothelial cells contribute to a maximum of 6.99% of the total cells in any given sample, with an average 0.831% prevalence across samples. For both cell types, DWLS is able to maintain high estimation accuracy ($\rho_{\text{dendritic,DWLS}} = 0.93$, $\rho_{\text{endothelial,DWLS}} = 0.81$), outperforming ν-SVR ($\rho_{\text{dendritic,SVR}} = 0.91$, $\rho_{\text{endothelial,SVR}} = 0.54$), and QP ($\rho_{\text{dendritic,QP}} = 0.66$, $\rho_{\text{endothelial,QP}} = 0.44$) (Fig. 2b). Overall, these analyses indicate that DWLS exhibits greater accuracy in estimating rare cell types than existing methods.

**DWLS extends to real bulk data characterized by the MCA.** Recently, Han et al. have characterized 43 healthy mouse tissues at single-cell resolution to create the Mouse Cell Atlas[18]. Based on a combined single-cell data set of 61k cells, they have identified 52 distinct cell types spread across all tissues. Here we selected four represented tissues—kidney, lung, liver, and small intestine—and generated two bulk RNA-seq data sets per tissue. Obtaining both bulk and single-cell data from the same tissue provides an opportunity to rigorously evaluate the accuracy of our deconvolution method, where we assume cell-type composition in bulk and single-cell data sets to be approximately equal. We use the entire single-cell data set to provide a comprehensive gene expression signature.

We calculate estimates using various deconvolution methods: DWLS, ν-SVR, and QP. Overall, we find a high replicability of our results within each pair of tissues, each of which come from separate mice. DWLS estimates for each pair have correlations between 0.84 and 0.99, showing that cell-type composition differences between mice are small.

Here, DWLS again performs favorably over other methods, which we demonstrate in two ways. We first look at a representative example, the deconvolution of bulk kidney data (Fig. 3a, b). We plot deconvolution estimates against the predicted true cell-type composition, and find that DWLS estimates are most highly correlated to the predicted true proportion ($\rho_{\text{kidney,DWLS}} = 0.89$), with ν-SVR and QP performing

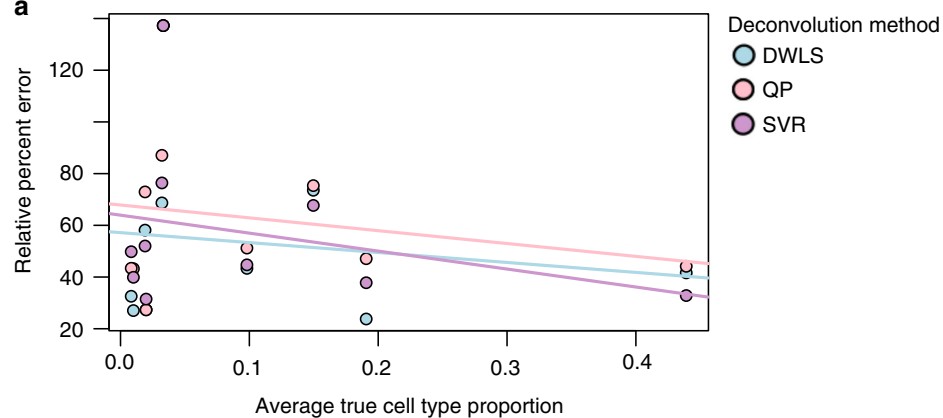

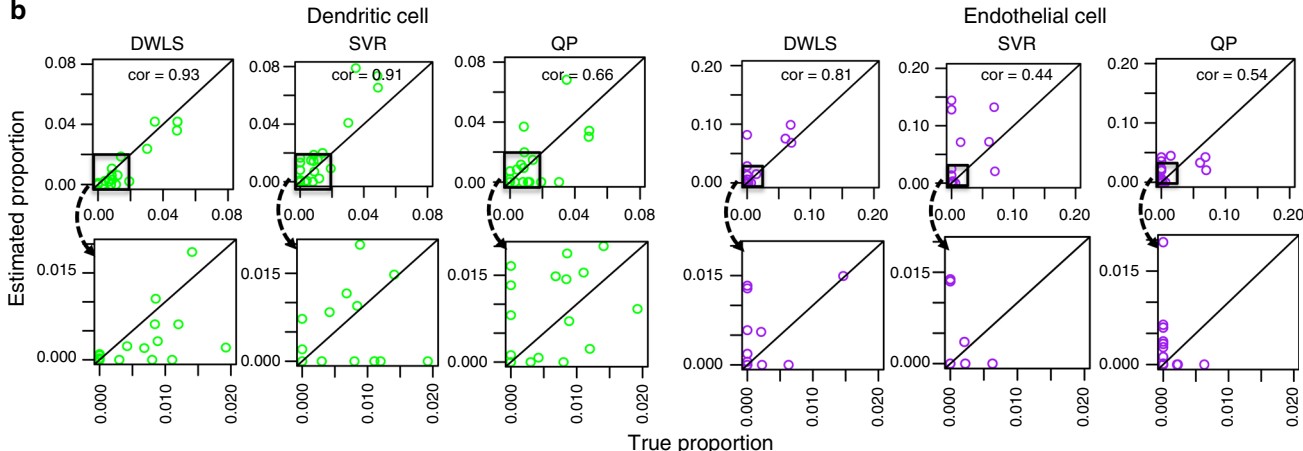

**Fig. 2** Results from the deconvolution of 27 simulated bulk data sets. **a** The mean relative percent error in estimation for each cell type across 27 simulated data sets from donor, melanoma, and ovarian cancer patient immune and tumor cells, plotted against the average true proportion of the cell type, for each method (dampened weighted least squares (DWLS), quadratic programming (QP), and $\nu$-support vector regression ($\nu$-SVR). The fitted lines represent the trend in estimation accuracy as a function of cell-type proportion. **b** A subset of the deconvolution cell-type proportion estimates, plotted against the true cell-type proportions. Here, only the rarest cell types, dendritic and endothelial cells, are shown. Correlation values between true and estimated proportions are used to quantify estimation accuracy. The 45° line in each plot represents the optimal estimate. The top row shows all estimates, while the bottom row shows a zoomed-in version focused on only the rarest cell types

less favorably ($\rho_{\text{kidney,SVR}} = 0.87$, $\rho_{\text{kidney,QP}} = 0.092$) (Fig. 3a). DWLS is the only method able to correctly predict the presence of all four kidney cell types. QP misses three out of these four groups entirely, while $\nu$-SVR misses one (Fig. 3b). $\nu$-SVR also significantly overestimates the presence of other rarer cell types (Fig. 3b), which should make up around 6% of the total kidney cell population, but are estimated by $\nu$-SVR to make up 43% instead.

Second, we look more generally at the estimates of all eight tissue samples analyzed. DWLS remains the most accurate method, with an average correlation of 0.78 for DWLS, compared with average correlations of 0.21 and 0.59 for QP and $\nu$-SVR, respectively (Fig. 3c). QP once again fails to detect biologically relevant cell types across the eight bulk samples. This can be quantified by a sensitivity metric, defined as the fraction of all true cell types that are detected by the deconvolution method. Across the eight bulk samples, QP deconvolution results are characterized by a low sensitivity (Fig. 3c). $\nu$-SVR once again erroneously predicts the presence of cell types that are known to be biologically irrelevant to the given tissue. This is measured using a specificity metric, defined as the fraction of all false cell types that are correctly undetected by the deconvolution method. Across the eight bulk samples, $\nu$-SVR deconvolution results are characterized by a low specificity

(Fig. 3c). Overall, DWLS strikes the best balance between these two metrics by being able to both detect correct cell types and ignore false cell types (Fig. 3c).

**DWLS captures ISC composition changes across conditions.** One of the most important applications of deconvolution methods is in the identification of cell-type composition variations across conditions. To test the utility of our deconvolution method, we turned to a public data set where mouse ISC compartments are perturbed by drug treatments. In particular, Yan et al.[19] explored the effects of R-spondin ligand (RSPO1-4) inhibition and gain-of-function on intestinal stem-cell regeneration and differentiation through bulk gene expression profiling. Since bulk RNA-seq analysis alone does not provide information regarding cell-type composition, they followed up with single-cell RNA-seq analysis and observed dramatic changes of cell-type composition in four distinct cell-type compartments: non-cycling ISC, cycling ISC, transit amplifying (TA), and differentiated cells. Here we use this data set to test whether our deconvolution method can reveal such changes based on bulk RNA-seq data alone.

We applied DWLS to estimate the cell-type composition changes due to these drug treatments, using the single-cell data only to estimate the cell-type-specific gene expression signature

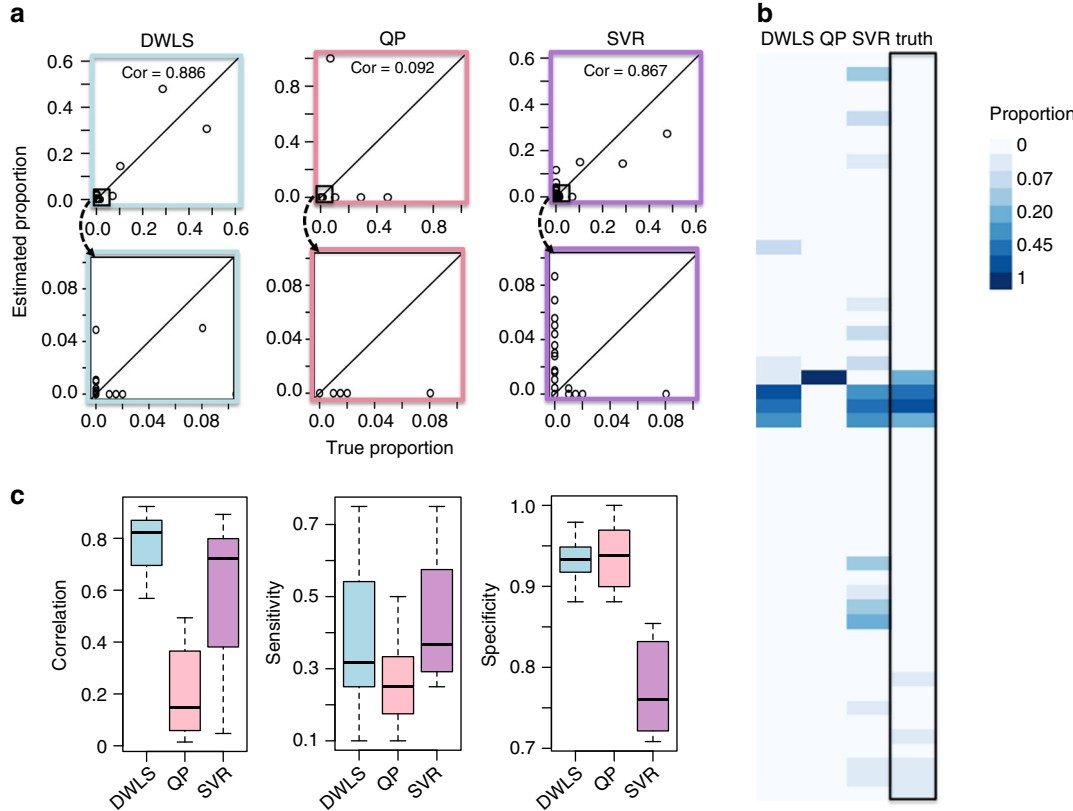

**Fig. 3** Deconvolution of eight normal mouse bulk data sets characterized by the MCA. **a** Results from the deconvolution of each bulk data set using a signature constructed from the mouse cell atlas (MCA), using three deconvolution methods: dampened weighted least squares (DWLS), quadratic programming (QP), and ν-support vector regression (ν-SVR). Estimates are plotted against an approximate true cell-type proportion as defined by the MCA data. Correlation values between true and estimated proportions are used to quantify estimation accuracy for each method. The 45° line in each plot represents the optimal estimate. The top row shows all estimates, while the bottom row shows a zoomed-in version focused on only the rarest cell types. **b** Another view of the kidney deconvolution estimates under each deconvolution method via a heatmap, where each box corresponds to a cell-type proportion estimate, and a darker color corresponds a higher estimated proportion. Colors are shown on a log scale. **c** A summary of deconvolution results across all eight bulk samples, quantified by (1) correlation between true and estimated cell-type proportions for each tissue (left panel), (2) sensitivity of each deconvolution method (middle panel), and (3) specificity of each deconvolution method (right panel). The center line of the boxplot corresponds to the median value, while bounds of the boxplot correspond to the 25th and 75th percentiles. The upper whisker bound corresponds to the smaller of the maximum value and the 75th percentile plus 1.5 interquartile ranges; lower corresponds to the larger of the smallest value and the 25th percentile minus 1.5 interquartile ranges

matrix (Fig. 4). We found that treatment with Ad-LGR5-ECD almost entirely removed the intestinal stem-cell population (on average, from 53.3 to 1.76%), while increasing the proportion of transit amplifying cells by 2.07-fold (25.5 to 52.8%) on average and differentiated cell types by 2.15-fold (21.1 to 45.4%) on average. On the other hand, treatment with Ad-RSPO1 completely removed the transit amplifying cell population, while increasing the size of the intestinal stem-cell population by an average 1.50-fold (53.3 to 79.8%). These observations are highly consistent with the single-cell RNA-seq data, which were used to deduce the biological functions of these treatments. That is, Ad-LGR5-ECD treatment drives differentiation, while Ad-RSPO1-treatment promotes stem-cell renewal. Here, we were able to draw the same conclusions without the need to generate single-cell RNA-seq data from every condition.

In comparison, inconsistencies arose when estimation was performed using QP and ν-SVR approaches. Specifically, neither method was consistently able to detect any cycling intestinal stem cells, whose proportion was estimated to be 29% in the control condition and 44% in the Ad-RSPO1 condition based on the single-cell RNA-seq data, and on average 31.8% and 31.4% according to the DWLS estimates. ν-SVR also predicted an increase in differentiated cell types due to Ad-RSPO1 treatment

(7.64 to 45.2%), which is inconsistent with the results of the other estimation methods, the single-cell RNA-seq data, and the underlying biological mechanisms[19].

**Model robustness evaluation.** To test the robustness of these results, we repeated the above ISC analysis while varying a number of model parameters, including normalization procedure, criteria for signature gene identification, and dampening magnitude, as described below.

First, single-cell RNA-seq data sets usually contain many dropouts due to amplification bias or other technical artifacts. To evaluate the effect of dropouts on deconvolution, we simulated single-cell RNA-seq data using Splatter[20], varying the dropout rate from 16% to 41% (see Methods section for details). For comparison, we also applied QP and ν-SVR to the same data sets. The accuracy of DWLS remains high across dropout rates (Supplementary Fig. 2). Importantly, DWLS is more robust than the other two methods, with the greatest difference between methods seen at high dropout rates.

Second, technical differences between single-cell and bulk RNA-seq data may induce significant biases on cell-type composition estimation. A number of methods have been

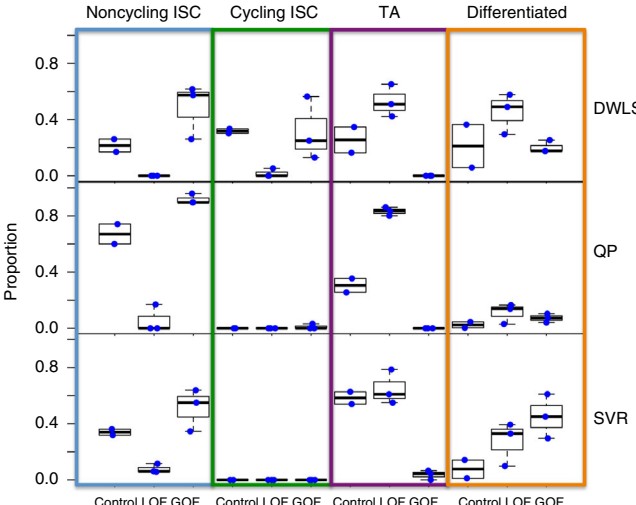

**Fig. 4** Deconvolution estimates of bulk mouse ISC data across various conditions. The control condition corresponds to Lgr5-eGFP+ intestine cells 1.5 days post treatment with Ad-Fc, the loss of function (LOF) condition corresponds to Lgr5-eGFP+ intestine cells 1.5 days post treatment with Ad-LGR5-ECD, and the gain of function (GOF) condition corresponds to Lgr5-eGFP+ intestine cells 1.5 days post treatment with Ad-RSPO1. Each point corresponds to the deconvolution estimate of a cell type for a single bulk data set, for the dampened weighted least squares (DWLS), quadratic programming (QP), and ν-support vector regression (ν-SVR) deconvolution methods. Cell types include cycling and non-cycling intestinal stem cells (ISCs), transit amplifying (TA) cells, and various differentiated cell types. The center line of the boxplot corresponds to the median value, while bounds of the boxplot correspond to the 25th and 75th percentiles. The upper whisker bound corresponds to the smaller of the maximum value and the 75th percentile plus 1.5 interquartile ranges; the lower corresponds to the larger of the smallest value and the 25th percentile minus 1.5 interquartile ranges

developed to normalize single-cell RNA-seq data[21,22,23]. To evaluate the effect of using different normalization methods on deconvolution, we used Splatter to simulate scRNA-seq data sets while introducing library size variation. We normalized the data using three different methods: Scran[21], Seurat[22], and SCnorm[23], and applied DWLS to analyze the normalized data (see Methods section for details). For comparison, we also used QP and ν-SVR to analyze the same data sets. We noticed significant differences across the different normalization methods, with SCnorm most faithfully preserving cell-type composition (Supplementary Fig. 3a, b, c). Of note, while SCnorm normalizes UMI counts directly, both Scran and Seurat log-transfrom the data after normalization. To test if the performance difference between normalization methods is due to log-transformation, we reanalyzed the data by applying Scran and Seurat without log-transformation. In both cases, the performance is much improved (Supplementary Fig. 3a, b, c).

While all three deconvolution methods perform well in all normalization methods without log-transformation, the performance of DWLS is slightly better in this case. To further evaluate the effect of normalization in real data analysis, we reanalyzed the ISC data set described above after normalization. In this case, the DWLS results are robust with respect to different normalization methods, whereas QP and ν-SVR are much more sensitive (Supplementary Fig. 3d, e, f). Importantly, only DWLS recapitulates the correct trend due to treatment. Again, the difference between normalization methods is mainly due to log-transformation (Supplementary Fig. 3g).

Third, we evaluated the robustness of DWLS to the selection of signature genes. To this end, we used a more stringent cutoff (log (fold change) ≥ 1 and $p$-value < 0.001, FDR adjusted, defined using the hurdle model in the MAST R package) to select differentially expressed genes. As a result, the number of differentially expressed genes was reduced from 1032 to 775. Using this more stringent gene signature, we re-estimated cell-type composition using DWLS. We found the results were similar to before (Supplementary Fig. 4). Furthermore, we compared the signature genes obtained from two different differential gene detection methods: Voom[24] and edgeR[25]. In both cases, we found that the results were similar to before, in part due to the strong overlap of the detected gene signature from different methods (Supplementary Fig. 4).

## Discussion

Cellular heterogeneity must be taken into account when comparing gene expression data from bulk samples. As large efforts are under way to thoroughly characterize cell types of different organisms through single-cell analyses[26], we are facing a new opportunity to systematically quantify cell-type composition using the detected cell-type signatures. We envision that such deconvolution methods will be routinely used to precisely determine gene expression pattern changes in development and disease. Toward this goal, we have developed a new and more accurate computational method for deconvolution.

Using the mouse cell atlas data set as an example, we have demonstrated that the tissue of origin of a bulk sample can be accurately predicted from deconvolution given a comprehensive signature of all cell types in an organism. In the meantime, we also recognize the danger of detecting irrelevant cell types, which is especially acute when many irrelevant cell types are included in the signature. Cell types from different tissues may share similar functions and therefore may be difficult to differentiate due to high collinearity. To minimize this risk, we advise that after a general deconvolution with a broad signature, irrelevant cell types be removed from the signature matrix to build a more specific signature matrix from only the most appropriate single-cell data sets. Such a multi-step approach may result in both more specific cell-type designations and more accurate estimates, although further investigation is needed to validate this approach.

At the other end of the spectrum, deconvolution accuracy is always dependent on the completeness of the cell-type signature, and incomplete cell-type information will compromise estimates of all cell types in the signature. Care must always be taken to create the most appropriate signature matrix given the extent of information known about the sample. Overall, the flexibility of `signature matrix definitions made possible by large quantities of single-cell data has promising implications.

Another challenge in deconvolution is the accurate estimation of rare cell types. In part, this is because detecting rare cell types from a large population in single-cell data is a challenging task, and precise signatures are difficult to build[27–30]. In addition, the estimation of rare cell proportions by deconvolution is notoriously difficult due to the increased stochasticity of small sample sizes[2]. While our method presents an improvement over previous methods in rare cell-type detection, we hope to further improve rare cell-type detection accuracy in future work.

## Methods

**Implementation**. The DWLS method is implemented in a Cran R package called DWLS. Both source codes and instructions are available at https://bitbucket.org/yuanlab/dwls.

**Creation of the signature matrix.** The cell-type signature matrix is constructed using a representative single-cell data set, such that all cell types expected in the bulk data are also represented in the single-cell data (the converse need not be true). The single-cell data is first clustered to reveal its constituent cell types. The optimal clustering method is dependent on the data set, but generally, a rare-cell-type-sensitive clustering method is preferred[27–30]. Further inspection of differentially expressed genes between each of these clusters is important, as this will confirm whether the detected clusters consist of biologically relevant cell types. Upon characterization of the cell types, differential expression analysis is performed to identify marker genes for each cell type. We define marker genes as genes with an FDR adjusted $p$-value of <0.01 (defined using the hurdle model in the MAST R package), and a log2 mean fold change >0.5. For very large single-cell data sets like the Mouse Cell Atlas, $p$-values are instead determined using the Seurat R package under the bimod likelihood ratio test for single-cell gene expression[31], due to the faster runtime. To create the final signature matrix $\mathbf{S}$, we create many candidate matrices (151 in total), which include between 50 and 200 marker genes from each cell type. The expression values of these chosen genes are averaged across each cell type, so that each resulting candidate matrix is an $n \times k$ matrix, where $n$ is the number of genes and $k$ is the number of cell types. The final signature matrix $\mathbf{S}$ is chosen as the candidate matrix with the lowest condition number, in a manner similar to CIBERSORT[9].

**Mathematical details of our weighted least squares approach.** To be more precise, we rewrite the deconvolution problem as $\hat{\mathbf{S}}\hat{\mathbf{x}} = \mathbf{t}$, where $\hat{\mathbf{S}}$ is the signature matrix derived above, $\hat{\mathbf{x}}$ is the estimated cell-type number, and $\mathbf{t}$ is the bulk data. Most notably, $\hat{\mathbf{S}}$ is used to denote that the single-cell-derived signature is only an estimate of the true cell-type signature, $\mathbf{S}$, which is unknown. Similarly, $\hat{\mathbf{x}}$ is the solution to $\hat{\mathbf{S}}\hat{\mathbf{x}} = \mathbf{t}$, which will almost always differ from the true cell-type number, $\mathbf{x}$, which is only known in the case of simulated bulk data. Suppose we have $k$ cell types and $n$ signature genes. Let $\mathbf{t} = (t_1 t_2 \dots t_n)'$, $\hat{x} = (\hat{x}_1 \hat{x}_2 \dots \hat{x}_k)'$, and

$$\hat{\mathbf{S}} = \begin{bmatrix} \hat{S}_{11} & \dots & \hat{S}_{1k} \\ \vdots & \ddots & \vdots \\ \hat{S}_{n1} & \dots & \hat{S}_{nk} \end{bmatrix}.$$ This system of equations can be solved in various ways.

In the traditional setting, we obtain an estimate, $\hat{\mathbf{x}}$, of the true cell type $\mathbf{x}$ by minimizing the squared error:

$$\hat{\mathbf{x}} = \underset{\tilde{\mathbf{x}}}{\text{argmin}}\, \text{Err}(\mathbf{t}, \hat{\mathbf{S}}, \tilde{\mathbf{x}}) = \underset{\tilde{\mathbf{x}}}{\text{argmin}} \sum_{i=1}^{n} \left( t_i - \sum_{j=1}^{k} \hat{S}_{ij} \tilde{x}_j \right)^2 \quad (1)$$

Define $\tilde{x}_j = \frac{x_j}{x_1} \tilde{x}_1$, for $j = 2, \dots, k$. Then,

$$\begin{aligned} \text{Err} &= \sum_{i=1}^{n} \left( t_i - \hat{S}_{i1} \tilde{x}_1 - \sum_{j=2}^{k} \hat{S}_{ij} \frac{x_j}{x_1} \tilde{x}_1 \right)^2 \\ &= \sum_{i=1}^{n} \left( t_i - \hat{S}_{i1} \tilde{x}_1 \hat{q}_i \right)^2, \text{ where } \hat{q}_i = \left( 1 + \sum_{j=2}^{k} \frac{x_j}{x_1} \frac{\hat{S}_{ij}}{\hat{S}_{i1}} \right) \\ &= \sum_{i=1}^{n} \left( S_{i1} x_1 q_i - \hat{S}_{i1} \tilde{x}_1 \hat{q}_i \right)^2, \text{ where } q_i = \left( 1 + \sum_{j=2}^{k} \frac{x_j}{x_1} \frac{S_{ij}}{S_{i1}} \right) \\ &= \sum_{i=1}^{n} \left( S_{i1} x_1 q_i - \hat{S}_{i1} x_1 \hat{q}_i + \hat{S}_{i1} x_1 \hat{q}_i - \hat{S}_{i1} \tilde{x}_1 \hat{q}_i \right)^2 \end{aligned} \quad (2)$$

To proceed, we make an assumption that the cross terms are orthogonal to each other, i.e.,

$$\sum_{i=1}^{n} \left( \left( S_{i1} x_1 q_i - \hat{S}_{i1} x_1 \hat{q}_i \right) \left( \hat{S}_{i1} x_1 \hat{q}_i - \hat{S}_{i1} \tilde{x}_1 \hat{q}_i \right) \right) = 0 \quad (3)$$

This is equivalent to assuming that the estimation error for cell-type signatures is independent of the estimation error for cell-type composition. While this assumption cannot be rigorously tested in real data, an approximate test of this assumption is possible through bootstrapping methods. With this assumption, Eq. (2) can be further reduced to

$$\text{Err} = \sum_{i=1}^{n} \left( S_{i1} x_1 q_i - \hat{S}_{i1} x_1 \hat{q}_i \right)^2 + \sum_{i=1}^{n} \left( \hat{S}_{i1} x_1 \hat{q}_i - \hat{S}_{i1} \tilde{x}_1 \hat{q}_i \right)^2 \quad (4)$$

The first term is driven by a difference between the true and estimated signatures, which cannot be controlled for. We concern ourselves with the second term, which can be rewritten as:

$$\sum_{i=1}^{n} \left( \hat{S}_{i1} \hat{q}_i (x_1 - \tilde{x}_1) \right)^2 = \sum_{i=1}^{n} \left( \sum_{j=1}^{k} \hat{S}_{ij} x_j \right)^2 \left( \frac{(x_1 - \tilde{x}_1)}{x_1} \right)^2 \quad (5)$$

We can see that this error term corresponds to the relative error of estimation for cell type 1, multiplied by a function of $\hat{\mathbf{S}}$ and $\mathbf{x}$ such that genes with high expression and genes pertaining to prevalent cell types will have a larger impact on the error term. Because we would like all cell types to be estimated with equal accuracy, we would like the error term to be a function of the relative error of estimation only.

To mitigate this problem, we use a weighted least squares approach to solve the equation, which is represented as the following optimization:

$$\min_{\tilde{\mathbf{x}}} \sum_{i=1}^{n} w_i \left( t_i - \sum_{j=1}^{k} \hat{S}_{ij} \tilde{x}_j \right)^2 \quad (6)$$

The weights are chosen as to remove the extra term in Eq. (6). If we let:

$$w_i = \frac{1}{\left( \sum_{j=1}^{k} \hat{S}_{ij} x_j \right)^2} = \frac{1}{\left( \hat{\mathbf{S}}_{i\cdot} \mathbf{x} \right)^2} \quad (7)$$

we are now minimizing:

$$\begin{aligned} \text{Err} &= \sum_{i=1}^{n} w_i \left( t_i - \sum_{j=1}^{k} \hat{S}_{ij} \tilde{x}_j \right)^2 \\ &\approx \sum_{i=1}^{n} \frac{1}{(\hat{\mathbf{S}}_{i\cdot} \mathbf{x})^2} \left( \hat{\mathbf{S}}_{i\cdot} \mathbf{x} \right)^2 \left( \frac{(x_1 - \tilde{x}_1)}{x_1} \right)^2 = \sum_{i=1}^{n} \left( \frac{(x_1 - \tilde{x}_1)}{x_1} \right)^2 = n \left( \frac{(x_1 - \tilde{x}_1)}{x_1} \right)^2, \end{aligned} \quad (8)$$

such that the total error is now a function of the relative error in cell-type number for cell type 1. Without loss of generality, we can similarly show this relationship for any cell type $j \in \{1, \dots, k\}$, such that

$$\text{Err} \approx n \left( \frac{\left( x_j - \tilde{x}_j \right)}{x_j} \right)^2 \forall j \in \{1, \dots, k\} \quad (9)$$

Compared with the ordinary least squares approach, the relative error is reduced.

**Additional adjustments to improve performance.** Using the framework derived above, we would like to formulate the estimation of cell-type proportion as a weighted least squares problem with weights $w_i = \frac{1}{(\hat{\mathbf{S}}_{i\cdot} \mathbf{x})^2}$. Several modifications are required to make this a viable approach:

(1) The weights are a function of $\mathbf{x}$, the true cell-type number, which is unknown. We can approximate this with our estimated cell-type number, $\hat{\mathbf{x}}$, but since this also the variable being solved for, iteration is required to reach a solution. Let:

$$\hat{\mathbf{x}}^{(0)} = \underset{\tilde{\mathbf{x}}}{\text{argmin}} \sum_{i=1}^{n} \left( t_i - \sum_{j=1}^{k} \hat{S}_{ij} \tilde{x}_j \right)^2 \quad (10)$$

$$\hat{\mathbf{x}}^{(1)} = \underset{\tilde{\mathbf{x}}}{\text{argmin}} \sum_{i=1}^{n} w_i^{(1)} \left( t_i - \sum_{j=1}^{k} \hat{S}_{ij} \tilde{x}_j \right)^2, \text{ where } w_i^{(1)} = \frac{1}{\left( \hat{\mathbf{S}}_{i\cdot} \hat{\mathbf{x}}^{(0)} \right)^2} \quad (11)$$

...

$$\hat{\mathbf{x}}^{(l)} = \underset{\tilde{\mathbf{x}}}{\text{argmin}} \sum_{i=1}^{n} w_i^{(l)} \left( t_i - \sum_{j=1}^{k} \hat{S}_{ij} \tilde{x}_j \right)^2, \text{ where } w_i^{(l)} = \frac{1}{\left( \hat{\mathbf{S}}_{i\cdot} \hat{\mathbf{x}}^{(l-1)} \right)^2} \quad (12)$$

Convergence is reached when $\left\| \hat{\mathbf{x}}^{(l)} - \hat{\mathbf{x}}^{(l-1)} \right\| \leq 0.01$.

(2) The weights are unbounded from above and may approach infinity in the case of very rare cell types ($\hat{\mathbf{x}} \approx 0$) and/or lowly expressing genes ($\hat{S}_{ij} \approx 0$ for all cell types). This will lead to a solution driven by only a few genes. To rectify this, a dampening constant $d$ is introduced, which defines the maximum value any weight can take on. For ease of use, we first linearly scale the weights such that the minimum weight takes on a value of 1: $w_i^s = \frac{w_i}{\min(w_j)}, j \in \{1, \dots n\}$. The resulting optimization is equivalent. The dampened weights $\tilde{w}_i$ are then defined as:

$$\tilde{w}_i = \left\{ \begin{array}{l} w_i^s, \text{ if } w_i^s < d \\ d, \text{ otherwise} \end{array} \right\}$$

Cross-validation is used to select $d$, as follows. The possible values for $d$ are defined as $d = 2^q$, where $q \in \{0, 1, 2, \dots \max(\text{noninfinite} \log 2(w_i^s))\}$. Then, 100 subsets of signature genes of half the size of the full signature gene set are randomly selected. For each subset, the cell-type proportion is estimated using weighted least squares on the dampened weights, for each possible value of $d$. The variance of the estimates over the 100 subsets for each choice of $d$ is calculated, and the $d$ that leads to the lowest variance is selected. Alternatively, the cross-validation criterion may be changed to minimize the coefficient of variation instead of the variance. In practice, we find the results are often similar (Supplementary Fig. 5).

(3) As specified above, $\hat{\mathbf{x}}$ need not be positive. However, cell-type numbers are inherently non-negative. To set a constraint on $\hat{\mathbf{x}}$, such that $\hat{x}_j \geq 0\, \forall j$, we solve the constrained dampened weighted least squares problem via quadratic programming, using the function solve. QP in the R package

quadprog. The new minimization problem is then:

$$\min_{\tilde{\mathbf{x}}, \tilde{x} \geq 0} \sum_{i=1}^{n} \tilde{w}_i \left( t_i - \sum_{j=1}^{k} \hat{S}_{ij} \tilde{x}_j \right)^2$$

Jointly implementing all of these alterations, we reach the final deconvolution process:

$$\hat{\mathbf{x}}^{(0)} = \underset{\tilde{\mathbf{x}}}{\operatorname{argmin}} \sum_{i=1}^{n} \left( t_i - \sum_{j=1}^{k} \hat{S}_{ij} \tilde{x}_j \right)^2 \tag{13}$$

$$\hat{\mathbf{x}}^{(1)} = \underset{\tilde{\mathbf{x}}, \tilde{x} \geq 0}{\operatorname{argmin}} \sum_{i=1}^{n} \tilde{w}_i^{(1)} \left( t_i - \sum_{j=1}^{k} \hat{S}_{ij} \tilde{x}_j \right)^2, \text{ where } \tilde{w}_i^{(1)} = \operatorname{damp}\left( \frac{1}{(\hat{S}_{i,\cdot} \hat{x}^{(0)})^2} \right), \text{ and } \operatorname{damp}(w_i)$$

$$= \begin{cases} \frac{w_i}{\min(w_j)}, & \text{if } \frac{w_i}{\min(w_j)} < d \\ d, & \text{otherwise} \end{cases} \tag{14}$$

$$\cdots$$

$$\hat{\mathbf{x}}^{(l)} = \underset{\tilde{\mathbf{x}}, \tilde{x} \geq 0}{\operatorname{argmin}} \sum_{i=1}^{n} w_i^{(l)} \left( t_i - \sum_{j=1}^{k} \hat{S}_{ij} \tilde{x}_j \right)^2, \text{ where } \tilde{w}_i^{(l)} = \operatorname{damp}\left( \frac{1}{\left( \hat{S}_{i,\cdot} \hat{x}^{(l-1)} \right)^2} \right) \tag{15}$$

Convergence is reached when $\left\| \hat{\mathbf{x}}^{(l)} - \hat{\mathbf{x}}^{(l-1)} \right\| \leq 0.01$.

**Simulation details**. Counts for the simulated single-cell data set are generated using a Poisson distribution, for a total of six genes and three cell types. In the first simulation, two genes are upregulated in each cell type, where $\lambda = 50$ for an upregulated gene and $\lambda = 5$ otherwise. Fifty cells from each cell type are used to create a signature matrix, where the six genes are averaged over each cell type to create a reference gene expression profile. Between 10,001 and 15,000 cells are used to simulate bulk data, by summing up gene expression values across cell types. Specifically, 5000 bulk data sets are created by combining 5000 cells from cell types 2 and 3, and between 1 and 5000 cells from cell type 1. Overall, this creates bulk data sets with a rare cell-type proportion spanning between 0.1% and 33.3%. Bulk data simulation is repeated 10 times for each rare cell-type proportion, and all metrics reported are based on an average of these 10 samples.

In the second simulation, two genes are again upregulated in each cell type, but the mean expression level of the marker genes corresponding to the first cell type is lower, such that $\lambda$ ranges from 0.05 to 10 for an upregulated gene and from 0.005 to 1 otherwise. Fifty cells from each cell type are again used to create a signature matrix, where gene expression levels are scaled for each choice of $\lambda$, for a total of 200 signature matrices. To simulate the bulk data, 5000 cells from each cell type are aggregated so that each cell type is present in equal proportion. Bulk data simulation is repeated 10 times for each choice of $\lambda$, and all metrics reported are based on an average of these 10 samples.

**Estimation using other deconvolution methods**. Nu-support vector regression was performed using the svm function in the e1071 package in R. Parameters were set to nu = 0.5, type = "nu-regression", kernel = "linear", cost = 1, and all others to the default values. Bulk data and signature matrices were scaled to $[-1, 1]$. These parameter and scaling choices match those specified in Schelker et al.[17] in their MATLAB code, accessed through https://figshare.com/s/865e694ad06d5857db4b. As in Newman et al.[9], model coefficients are extracted from the svm model using t(model$coefs) %*% model$SV, and any negative coefficients are set to zero. The coefficients are then scaled by the sum of the coefficients, such that the scaled coefficients will sum to one.

Quadratic programming is implemented using the solve.QP function in the quadprog package in R. Default parameters are used, and the constraints are specified such that all coefficients must be greater than or equal to zero.

**MCA Bulk RNA-seq data pipeline**. Six- to ten-week-old male C57BL/6J mice were purchased from the Shanghai Laboratory Animal Center (SLAC). From each mouse, four non-sexual tissues (liver, small intestine, lung, and kidney) were excised. The excised tissues were immediately washed in PBS. After washing, each tissue was ground into powder with liquid N2. RNA extraction was performed using Trizol. We used mRNA Capture Beads (VAHTS mRNA-seq v2 Library Prep Kit for Illumina, Vazyme) to extract mRNA from total RNA. A PrimeScript Double Strand cDNA Synthesis Kit (TaKaRa) was used to synthesize double-stranded cDNA from purified polyadenylated mRNA templates according to the manufacturer's protocol. We used TruePrep DNA Library Prep Kit V2 for Illumina (Vazyme) to prepare cDNA libraries for Illumina sequencing (VeritasGenetics). All experiments performed in this study were approved by the Animal Ethics Committee of Zhejiang University. All experiments conform to the relevant regulatory standards at Zhejiang University Laboratory Animal Center.

Bulk sequencing reads containing multiplexed data were filtered using the bbduk function of the bbmap tool to select reads containing the appropriate sample index. STAR 2.5.3a[32] with default parameters was used to map filtered reads to the

Ensembl release 75 mouse reference genome. Aligned reads were normalized by library size to fragments per kilobase of transcript per million mapped reads (FPKM) using the fpkm function in the DESeq2 package in R.

**Mouse cell atlas single-cell data**. The mouse cell atlas (MCA) single-cell data[18] and annotations were accessed through https://figshare.com/s/865e694ad06d5857db4b. The single-cell data is quantified as UMI counts. The signature matrix was built using the 61k cell subset consisting of randomly sampled cells from 43 tissues. Cell types were defined by collapsing the 98 clusters identified by Han et al. into 52 unique cell types.

**Intestinal stem-cell bulk and single-cell data**. Intestinal stem-cell (ISC) single-cell and bulk RNA-seq data sets from Yan et al.[19] were accessed through the Gene Expression Omnibus (GEO) repository under accession numbers GSE92865 and GSE92377, respectively. The single-cell data is quantified as UMI counts. All Lgr5-eGFP+ and Lgr5-eGFP− cells were used in the construction of the signature matrix. The single-cell data cell-type labels shown in Yan et al. Figure 5a[19] were obtained from the authors upon request, and these were used to generate the signature matrix. The bulk data is quantified in terms of FPKM values.

**Schelker et al. simulation details**. Source code and data from the Schelker et al.[17] simulation analysis was accessed through https://figshare.com/s/711d3fb2bd3288c8483a. The single-cell RNA-seq data used in Schelker et al.[17] includes tumor cells from 19 melanoma patients, PBMCs from four healthy subjects, and ascite samples from four ovarian cancer patients. A signature matrix was built using all cells, using the clusters found by DBSCAN in Schelker et al.[17], and using the genes from the CIBERSORT immune-cell signature[9]. Twenty-seven patient-specific simulated bulk data sets were built by summing up gene expression values of signature genes across all cell types, for each patient.

**Modified relative percent error calculation**. Modified relative percent error measures the absolute difference between estimated and true cell-type proportions, normalized by the mean of the estimated and true cell-type proportions. A pseudo count of 0.005 is added so that for very small cell-type proportions, relative error does not become unreasonably high. It is defined as:

$$\text{MRPE} = \begin{cases} 0, & \text{if } x_l = 0 \text{ and } \hat{x}_l = 0 \\ \dfrac{\left| \dfrac{x_l}{\sum_{j=1}^{k} x_j} - \dfrac{\hat{x}_l}{\sum_{j=1}^{k} \hat{x}_j} \right|}{\dfrac{\frac{x_l}{\sum_{j=1}^{k} x_j} + \frac{\hat{x}_l}{\sum_{j=1}^{k} \hat{x}_j}}{2} + 0.005} * 100, & \text{otherwise} \end{cases} \tag{16}$$

where $x_l$ and $\hat{x}_l$ are the true and estimated cell-type numbers, respectively, for cell type $l$.

**Model robustness evaluation**. All evaluation analyses were carried out for the ISC data sets. To generate single-cell data with dropout effects, the splatSimulate function from the Splatter R package was used with parameters group.prob = c (0.25, 0.25, 0.25, 0.25), nGenes = 10,000, batchCells = 1000, and method = "groups"[20], with various dropout.shape values (dropout.shape = −2000, −2, −1.3, −0.8, −0.6, and −0.3). Each group was simulated five times with different seeds (seed = 1, 2, 3, 4, and 5). Bulk gene expression data was simulated by aggregating all single cells. For comparison, DWLS, QP, and v-SVR were applied to analyze the same data sets.

To evaluate the effects of normalization on deconvolution, we first used the splatSimulate function from the Splatter R package to generate three simulated data sets (group.prob = c(0.90,0.10), nGenes = 10,000, batchCells = 1000, method = "groups") corresponding to three different cell-type composition scenarios. Bulk data were simulated as the sum of the individual single-cell data. For each simulated data set, the raw data were normalized using three different normalization methods: Seurat[22], Scran[21], and SCnorm[23], respectively, with the following settings: Seurat (normalization.method = "LogNormalize", scale.factor = 100000), Scran (centre_size_factors = TRUE) and SCnorm (K = 1, conditions = rep(c(1), each = 1000)). For each normalized data set, we applied DWLS, QP, and v-SVR to deconvolve cell-type composition as described in the previous section. Furthermore, we also evaluated the performance using the ISC data set as a representative example of real data, using the same analysis procedure with settings: Seurat (normalization.method = "LogNormalize", scale.factor = 100000), Scran (centre_size_factors = TRUE), and SCnorm (conditions = rep(c(1), each = 12449)). Both Seurat and Scran log-transform the data after normalization. To test if the log-transformation step significantly affects deconvolution accuracy, we analyzed the data with or without log-transformation. The latter was derived by converting the log-transformed data back to the original scale.

To evaluate the influence of the signature gene selection, approximately half of the signature genes were randomly selected 10 times in the ISC data set for further analysis. A more stringent cutoff of log(fold change) $\geq 1$ and $p$-value < 0.001 (FDR adjusted, defined using the hurdle model in the MAST R package for ISC data, and

the bimod likelihood ratio test in the Seurat R package for MCA) was used for signature gene selection in ISC and MCA data sets.

Furthermore, we compared our results using two alternative differential gene detection methods: Voom[24] and edgeR[25]. For Voom, genes with log(fold change) ≥ 1 and $p$-value < 0.001 (FDR adjusted, empirical Bayes moderation method in the limma R package) were selected. For edgeR (FDR adjusted, Fisher's exact test in the edgeR R package), highly significant (top 100, top 200, top 300, top 400, and top 500) differential genes were selected. In each case, the selection of differentially expressed genes was followed by DWLS.

**Reporting summary**. Further information on research design is available in the Nature Research Reporting Summary linked to this article.

## Data availability

MCA Bulk RNA-seq data have been deposited to the Gene Expression Omnibus (GEO) database with accession code GSE124419. All other relevant data is available upon request.

## Code availability

Source code is freely accessible to the public at: https://bitbucket.org/yuanlab/dwls/src/default/

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

## Acknowledgements

We thank Grace Zheng for help procuring information for the intestinal stem-cell analysis, and Semir Beyaz for his insightful suggestions regarding method development and validation. We would also like to thank all members of the Yuan lab and Drs. John Quackenbush and Martin Aryee for their continued support.

## Author contributions

D.T. and G.C.Y. designed the computational method. D.T. and R.D. implemented the method. H.C. generated the bulk data for the Mouse Cell Atlas analysis under the supervision of G.G. Q.Z. created the R package. D.T. and G.C.Y. wrote the paper. All authors participated in the revision of the final paper.

## Additional information

**Competing interests:** The authors declare no competing interests.

