## [Peer Review File · Nature Communications]

Reviewers' comments:

Reviewer #1 (Remarks to the Author):

This work proposes a modification to the standard ordinary least square method, which they called dampened weighted least squares (DWLS), to perform cell-type deconvolution of bulk RNA-seq data, using a scRNA-seq data as a reference. This paper demonstrated that main difficulty of using scRNA-seq data as a reference is there is likely biased against rare cell-types and genes with low expression. Their paper shows that DWLS can largely alleviate both of these problems. The method is sensible, and the results look promising.

Nonetheless, I have the following concerns:

1. scRNA-seq is known to have a high level of dropouts, and the dropout rate is highly dependent on gene expression levels. By considering genes with low expression levels, wouldn't this inadvertently select more genes with higher dropout rate, and therefore introduce more noise?
2. Can the authors discuss the impact of normalisation (of scRNA-seq and bulk RNA-seq) and clustering of scRNA-seq data?
3. Can DWLS be provided as an easy-to-install R package (via bioconductor or similar)? It seems a bit strange that I need to open a Microsoft Word file to read the instruction to run the R script in the github page.

Reviewer #2 (Remarks to the Author):

Tsoucas et al. propose a new deconvolution strategy for estimating cell type proportions in bulk RNA-seq data based on reference single-cell expression profiles. In particular, their method focuses on improving estimation accuracy for rare cell types. They test their method on simulated data sets to demonstrate the improved accuracy, and on real data to show that it outperforms existing methods for rare cells. The manuscript is concise and addresses an aspect of the deconvolution problem that is generally overlooked. However, I have a number of concerns with the statistical justifications underlying the proposed method. I provide more details in my comments below.

MAJOR:

1. The iterative reweighting procedure lacks proof of convergence towards the global minimum. The weighted least-squares expression on page 19 contains, in effect, \tilde{x}_j in both the numerator and denominator. It is not clear to me that this expression has a single global minimum, though if it does, the authors should show that it exists. If such a minimum does not exist, the reweighting procedure runs the risk of being trapped in local minima. The authors should then provide practical advice for how to diagnose and avoid this effect. For example, are the results of the algorithm robust to different initial values of \tilde{x}_j ?
2. The authors assume orthogonality of the cross-terms on page 17 to get to the second expression on this page. This would require either $S_{i1}k_i = \hat{S}_{i1}\hat{k}_i$; or even worse, $x_{i1} = \tilde{x}_{i1}$, which cannot be true as otherwise the problem would already be solved! The first condition seems more reasonable and is an additional assumption that should be stated.
3. There is no mention of the differences between bulk and single-cell RNA sequencing protocols. For example, many single-cell RNA-seq protocols use unique molecular identifiers (UMIs) while most bulk

protocols do not. Other differences include the number of amplification cycles, where higher cycle numbers in single-cell data tend to favour shorter transcripts. This will probably affect the quality of deconvolution, as the single-cell expression values will be (mostly) independent of transcript length, while the bulk data are not. Such differences may lead to systematic biases in the estimated proportions, especially if the signature genes for a particular cell type are consistently longer/shorter than others. (This would not be a problem if the reasoning on page 17 were correct, but see point 2.) The authors should address this concern more specifically, e.g., by modelling single-cell vs bulk biases.

4. There is no statement of the error distribution of t . Presumably the authors are assuming that the errors are Normally distributed, given that they are using least-squares for fitting the model. It could even be said that the error is Normal with variance equal to the squared mean, which would (i) better reflect the mean-variance relationship in expression data; and (ii) better motivate the use of the mean in the weighting scheme on page 18.

5. The authors describe x_j as the number of cells of type j . However, if a cell type contains more RNA, fewer cells of that type are required to contribute the same expression to the bulk profile (compared to a cell type with less RNA). It seems that the authors have not considered the consequences of differences in RNA content between cell types. The construction of the signature matrix on page 15 is done using standard methods for scRNA-seq analysis, all of which involve normalization to remove differences in RNA content. Thus, x_j is more accurately described as a more abstract "relative contribution" of each cell type j , rather than as an absolute number of cells. If the authors still want to estimate absolute cell number, then they need to account for variation in RNA content across cell types in their construction of the signature matrix, e.g., by using spike-ins.

6. The authors cap the weights using a dampening scheme. The choice of the cap d is performed using "cross-validation", with the aim of choosing d that minimizes the variance of the results. However, there is no reason to think that the d with the lowest variance is correct. It is not uncommon for estimators to need to compromise between bias and precision, whereby increases to precision also yield a more biased estimator. It may well be that the choice of d that reduces variance also increases the bias in the proportion estimates. (One could, for example, trivially minimize the variance by setting all $x_j=0$, which would obviously be incorrect.) The authors should justify their choice of d more carefully.

7. How robust are the results to the number of signature genes? How robust are the results to the choice of signature genes? For example, are the same results obtained with different DE analysis methods (e.g., edgeR, DESeq2)? What is the variance of the proportions when subsets of the detected signature genes are used?

MINOR:

- The numbering of the colour scale in 3b is confusing.

- The results in Figures 2 and 3 could be presented much better. The aim of DWLS is to improve inference at low proportions, but all of the points around zero in Figures 2B and 3A are clustered together and difficult to resolve. It also suggests that the correlations are driven by a small handful of relatively high-abundance samples.

Point by point response to reviewers' comments.

We thank the reviewers for their thorough reading of our manuscript and constructive suggestions. Our point by point response to their comments is described below.

Reviewer #1 (Remarks to the Author):

This work proposes a modification to the standard ordinary least square method, which they called dampened weighted least squares (DWLS), to perform cell-type deconvolution of bulk RNA-seq data, using a scRNA-seq data as a reference. This paper demonstrated that main difficulty of using scRNA-seq data as a reference is there is likely biased against rare cell-types and genes with low expression. Their paper shows that DWLS can largely alleviate both of these problems. The method is sensible, and the results look promising.

Response: We thank the reviewer for his/her positive evaluation of our manuscript.

Nonetheless, I have the following concerns:

1. scRNA-seq is known to have a high level of dropouts, and the dropout rate is highly dependent on gene expression levels. By considering genes with low expression levels, wouldn't this inadvertently select more genes with higher dropout rate, and therefore introduce more noise?

Response: We appreciate the reviewer's concern. In order to evaluate the impact of dropout on deconvolution, we preprocessed the gene expression data by filtering out genes whose average UMI counts is less than 0.5 in every cell type, then applied our DWLS method to the filtered gene expression data. Using the ISC dataset as a representative example, we found that the estimated proportion of different cell-type was similar to our original results (Supp. Fig. S2). Thus, our deconvolution method is robust against drop-out effects.

2. Can the authors discuss the impact of normalisation (of scRNA-seq and bulk RNA-seq) and clustering of scRNA-seq data?

Response: Following the reviewer's suggestion, we evaluated the impact of normalization as follows. First, we normalized the single-cell RNAseq data by the total UMI reads in each cell. Then, we normalized the bulk and single-cell RNAseq data together by using quantile normalization. We applied DWLS to the normalized data and compared with our original results. As indicated in Supp Fig. 3, our results are robust against the specific choice of normalization methods.

3. Can DWLS be provided as an easy-to-install R package (via bioconductor or similar)? It seems a bit strange that I need to open a Microsoft Word file to read the instruction to run the R script in the github page.

Response: We thank the reviewer for this helpful suggestion and have created an easy-to-install R package. To run the package, a user simply needs to type the following commands:

```
library(devtools)
install_bitbucket("yuanlab/dwls", ref="default")
```

More details can be found at the package website:
<https://bitbucket.org/yuanlab/dwls>

Reviewer #2 (Remarks to the Author):

Tsoucas et al. propose a new deconvolution strategy for estimating cell type proportions in bulk RNA-seq data based on reference single-cell expression profiles. In particular, their method focuses on improving estimation accuracy for rare cell types. They test their method on simulated data sets to demonstrate the improved accuracy, and on real data to show that it outperforms existing methods for rare cells. The manuscript is concise and addresses an aspect of the deconvolution problem that is generally overlooked. However, I have a number of concerns with the statistical justifications underlying the proposed method. I provide more details in my comments below.

Response: We thank the reviewer for his/her positive evaluation of our manuscript.

MAJOR:

1. The iterative reweighting procedure lacks proof of convergence towards the global minimum. The weighted least-squares expression on page 19 contains, in effect, \tilde{x}_j in both the numerator and denominator. It is not clear to me that this expression has a single global minimum, though if it does, the authors should show that it exists. If such a minimum does not exist, the reweighting procedure runs the risk of being trapped in local minima. The authors should then provide practical advice for how to diagnose and avoid this effect. For example, are the results of the algorithm robust to different initial values of \tilde{x}_j ?

Response: We thank the reviewer for bringing up this important question. It is true that there is no theoretical guarantee that our method will converge to the global minimum. To ensure the solution is close to the global minimum, the user

can run our method multiple times, each starting from a different initialization, then choose the best solution. We tested this approach using the ISC dataset as a representative example. As indicated by Supp Fig. 1, the method converged for each initialization. Of note, the vast majority of initializations end up at the same result, although suboptimal solutions may result from a small subset of initial configurations. As such, from a practical point of view we feel it may not be necessary to rerun DWLS multiple times. The following discussion has been added in our revised manuscript:

“...Of note, while there is no theoretical guarantee that the converged solution reaches the global minimum, we find that in practice different initializations often end up at the same result, as demonstrated by our analysis of an intestine stem cell (ISC) dataset described later (Supp Fig. 1).”

2. The authors assume orthogonality of the cross-terms on page 17 to get to the second expression on this page. This would require either $\hat{S}_{i1}\hat{k}_i = \hat{S}_{i1}\hat{k}_i$; or even worse, $\hat{x}_1 = \tilde{x}_1$, which cannot be true as otherwise the problem would already be solved! The first condition seems more reasonable and is an additional assumption that should be stated.

Response: We apologize for the confusion. For simplicity, we did not show the exact formula corresponding to the orthogonality assumption, which is

$$\sum_{i=1}^n ((S_{i1}x_1k_i - \hat{S}_{i1}x_1\hat{k}_i)(\hat{S}_{i1}x_1\hat{k}_i - \hat{S}_{i1}\tilde{x}_1\hat{k}_i)) = 0$$

Note that this assumption does not require individual instances to be equal to zero, therefore it does not reduce to the extreme assumption that the reviewer derived. For clarification, we have added the above formula in the Methods section (page 18).

3. There is no mention of the differences between bulk and single-cell RNA sequencing protocols. For example, many single-cell RNA-seq protocols use unique molecular identifiers (UMIs) while most bulk protocols do not. Other differences include the number of amplification cycles, where higher cycle numbers in single-cell data tend to favour shorter transcripts. This will probably affect the quality of deconvolution, as the single-cell expression values will be (mostly) independent of transcript length, while the bulk data are not. Such differences may lead to systematic biases in the estimated proportions, especially if the signature genes for a particular cell type are consistently longer/shorter than others. (This would not be a problem if the reasoning on page 17 were correct, but see point 2.) The authors should address this concern more specifically, e.g., by modelling single-cell vs bulk biases.

Response: We thank the reviewer for bringing up this concern. We have carried out new analyses evaluating the impact of normalization using two different

methods including normalized UMI counts and quantile normalization. We found the results were robust against the specific normalization methods (Supp. Fig. 3).

4. There is no statement of the error distribution of t . Presumably the authors are assuming that the errors are Normally distributed, given that they are using least-squares for fitting the model. It could even be said that the error is Normal with variance equal to the squared mean, which would (i) better reflect the mean-variance relationship in expression data; and (ii) better motivate the use of the mean in the weighting scheme on page 18.

Response: The reviewer is correct. We indeed make the assumption that the errors approximately follow a normal distribution. To test this assumption, we have empirically examined the distribution of t . By visual inspection (see Figure below, each panel represents a different dataset), we find this assumption is valid.

5. The authors describe x_j as the number of cells of type j . However, if a cell type contains more RNA, fewer cells of that type are required to contribute the same expression to the bulk profile (compared to a cell type with less RNA). It seems that the authors have not considered the consequences of differences in RNA content between cell types. The construction of the signature matrix on page 15 is done using standard methods for scRNA-seq analysis, all of which involve normalization to remove differences in RNA content. Thus, x_j is more accurately described as a more abstract "relative contribution" of each cell type j , rather than as an absolute number of cells. If the authors still want to estimate absolute cell number, then they need to account for variation in RNA content across cell types in their construction of the signature matrix, e.g., by using spike-ins.

Response: We agree that the relative contribution of a cell-type can be either due to the number of cells or the amount of RNA contained in each cell. As in most studies, we make the assumption that the total amount of RNA from each cell is approximately equal. For clarification, we have added the following note on page 6 of the revised manuscript.

“Of note, we make the commonly used simplifying assumption that the total amount of RNA is approximately equal in each cell. If this is not true, the estimated contributions of each cell-type may not deviate from the actual cell abundance.”

6. The authors cap the weights using a dampening scheme. The choice of the cap d is performed using "cross-validation", with the aim of choosing d that minimizes the variance of the results. However, there is no reason to think that the d with the lowest variance is correct. It is not uncommon for estimators to need to compromise between bias and precision, whereby increases to precision also yield a more biased estimator. It may well be that the choice of d that reduces variance also increases the bias in the proportion estimates. (One could, for example, trivially minimize the variance by setting all $x_j=0$, which would obviously be incorrect.) The authors should justify their choice of d more carefully.

Response: We agree that both variance and bias are important factors in considering performance, as clearly demonstrated by the hypothetical, pathological scenario he/she proposed. To address this issue, we also considered an alternative criterion by minimizing the coefficient of variation, defined as the ratio of the standard deviation to the mean. We compared the results of these approaches using the ISC dataset as a representative example. We found that the results were quite similar (Supp. Fig 5). In our R package, we provide an option to use either approach to select the model parameter.

7. How robust are the results to the number of signature genes? How robust are the results to the choice of signature genes? For example, are the same results

obtained with different DE analysis methods (e.g., edgeR, DESeq2)? What is the variance of the proportions when subsets of the detected signature genes are used?

Response: We have tested the robustness of the results to the number of signature genes by varying the DE analysis methods as well as the significance cutoff, using the ISC dataset as the representative example. The results are indeed robust (Supp Fig. 4).

Of note, we were unable to test the DEseq2 method as the reviewer suggested, this is because DEseq2 is too time-consuming and, as a result, it failed to complete for the dataset we analyzed. As an alternative, we chose to compare with the edgeR and Voom methods instead.

MINOR:

- The numbering of the colour scale in 3b is confusing.

Response: We apologize for the confusion and have revised Fig 3b for clarification.

- The results in Figures 2 and 3 could be presented much better. The aim of DWLS is to improve inference at low proportions, but all of the points around zero in Figures 2B and 3A are clustered together and difficult to resolve. It also suggests that the correlations are driven by a small handful of relatively high-abundance samples.

Response: We apologize for the confusion and have revised Figs 2 and 3 for clarification.

Reviewers' comments:

Reviewer #1 (Remarks to the Author):

In my original review, I raised two concerns about the robustness of their method in relation to (1) dropouts and (2) normalisation method. The authors reported additional experimental results to address my concerns. Nonetheless, I do not believe their efforts addressed my concerns. Here are my reasons:

(1) Usually the main impact of dropouts in scRNA-seq analysis is not in genes with low expression across all samples. These samples would have contributed very little anyway. The main impact of dropouts is in those genes that a mixture of medium gene expression and zeros (possible dropouts). Their experiment of removing all genes that has low expression across all cell type did not address my concern. Furthermore, I do not necessarily agree with the authors's rebuttal that the 'results in Fig S2 was similar to our original results'. In Fig 4 (top row), DWLS estimates that in the control sample consists of roughly equal proportion (~25%) of noncycling ISC, cycling ISC, TA and differentiated cells. This is not similar to the results in Fig S2 which shows non-cycling ISC consists of 50% of the sample. In the case of GOF, Fig 4 shows that it consists of ~70% of non-cycling ISC, whereas Fig S2 shows that it consists of 50-60% of non-cycling ISC. Based on these data along, I don't think the authors are able to support their claim that their method is robust against dropouts. I suggest authors investigate this effect using simulation (where the amount of dropouts can be controlled), and allow apply their method on a variety of data that have different dropout rates.

(2) Similar to the above point, the experiments the authors added to test the robustness against normalisation was really superficial. To fully address this question, it is important to do a much more comprehensive analysis with several normalisation methods on real and simulation data. Furthermore, their results actually highlights DWLS may be sensitive to normalisation. In Fig 3B, DWLS shows that the LOF samples is ~90% differentiated cells, which is in big contrast to Fig 4, which suggests GOF samples should contains ~50% differentiated and 50% TA cells.

Reviewer #2 (Remarks to the Author):

The authors have addressed most of my concerns. I only have a few minor comments:

- It would be useful to provide a scientific/biological interpretation of the orthogonality assumption. Or in other words, how can a user judge if the assumption is appropriate for their data set?

- I am not sure why using the coefficient of variation is any better than using the variance for choosing "d". The former does not provide any further information on the potential bias/variance trade-off for a given choice of "d".

- The use of quantile normalization for single-cell data is concerning, especially if they are being coerced to the same distribution as the bulk data. It would be very courageous to assume that the true distribution of expression values in each single cell should have the same shape as that of the bulk sample; this assumption will be immediately violated by strong differential expression in a heterogeneous population. Perhaps the authors might consider more bespoke single-cell normalization approaches like scran and scNorm.

Point by point response to reviewers' comments.

We thank the reviewers for their thorough reading of our manuscript and constructive suggestions. Our point by point response to their comments is included below.

Reviewer #1 (Remarks to the Author):

In my original review, I raised two concerns about the robustness of their method in relation to (1) dropouts and (2) normalisation method. The authors reported additional experimental results to address my concerns. Nonetheless, I do not believe their efforts addressed my concerns. Here are my reasons:

(1) Usually the main impact of dropouts in scRNA-seq analysis is not in genes with low expression across all samples. These samples would have contributed very little anyway. The main impact of dropouts is in those genes that a mixture of medium gene expression and zeros (possible dropouts). Their experiment of removing all genes that has low expression across all cell type did not address my concern.

Response: We thank the reviewer for the clarification and agree that our previous analysis did not fully address his/her concerns. In this revision, we directly estimated the effect of dropout by using simulated data generated from Splatter (Zappia 2017). The advantage of using a simulated dataset is that the ground-truth is known. As shown in Supplementary Figure 2, the prediction accuracy remains high at least when dropout rate is less than 40%. Importantly, DWLS is more robust than existing methods at high dropout rates.

Furthermore, I do not necessarily agree with the authors's rebuttal that the 'results in Fig S2 was similar to our original results'. In Fig 4 (top row), DWLS estimates that in the control sample consists of roughly equal proportion (~25%) of noncycling ISC, cycling ISC, TA and differentiated cells. This is not similar to the results in Fig S2 which shows non-cycling ISC consists of 50% of the sample. In the case of GOF, Fig 4 shows that it consists of ~70% of non-cycling ISC, whereas Fig S2 shows that it consists of 50-60% of non-cycling ISC. Based on these data along, I don't think the authors are able to support their claim that their method is robust against dropouts. I suggest authors investigate this effect using simulation (where the amount of dropouts can be controlled), and allow apply their method on a variety of data that have different dropout rates.

Response: As discussed above, we agree that the approach we used previously is not appropriate for evaluating the effect of dropout on deconvolution. Following the

suggestion of the reviewer, we re-evaluated the robustness against dropout using simulated data where the ground-truth is known. We found that the accuracy of DWLS remains at high dropout rates and outperforms existing methods.

(2) Similar to the above point, the experiments the authors added to test the robustness against normalisation was really superficial. To fully address this question, it is important to do a much more comprehensive analysis with several normalisation methods on real and simulation data. Furthermore, their results actually highlights DWLS may be sensitive to normalisation. In Fig 3B, DWLS shows that the LOF samples is ~90% differentiated cells, which is in big contrast to Fig 4, which suggests GOF samples should contains ~50% differentiated and 50% TA cells.

Response: As suggested, we carried more thorough analyses to systematically evaluate robustness using three different methods (Seurat, Scran, and SCnorm) and tested on both simulated and real datasets. For simulated data, we also examined how accuracy changes with the population size. Our analysis shows that SCnorm significantly outperforms the other two normalization methods (Supplementary Figure 3a, b, c). To evaluate the effect on real data, we compared these methods on the ISC dataset. The DWLS results are robust with respect to different normalization methods, whereas QP and v-SVR are much more sensitive (Supplementary Figure 3d, e, f). Importantly, only DWLS recapitulates the correct trend due to treatment.

Reviewer #2 (Remarks to the Author):

The authors have addressed most of my concerns. I only have a few minor comments:

- It would be useful to provide a scientific/biological interpretation of the orthogonality assumption. Or in other words, how can a user judge if the assumption is appropriate for their data set?

Response: The first term, $S_{i1}x_1k_i - \hat{S}_{i1}x_1\hat{k}_i$, represents the estimation error for cell-type specific gene signatures, whereas the second term, $\hat{S}_{i1}x_1\hat{k}_i - \hat{S}_{i1}\tilde{x}_1\hat{k}_i$, represents the estimation error for cell-type proportions. Therefore, our basic assumption is that these two estimation errors are independent to each other. To test this assumption rigorously, one would need to know the ground-truth, which is clearly impossible for real data analysis. As an approximation, we could estimate the error distribution by bootstrapping the data. We have added a note in the revised Methods section to guide the readers.

- I am not sure why using the coefficient of variation is any better than using the variance for choosing "d". The former does not provide any further information on the potential bias/variance trade-off for a given choice of "d".

Response: The reason behind using the coefficient of variation is to emphasize the importance of accurately estimating the rare cell-type composition. If we just use the variance, even a large relative error may be unnoticed because the absolute error is likely to be small. On the other hand, we agree that this is not necessarily a better option than variance in some cases, and therefore provide both options to tune the parameter.

- The use of quantile normalization for single-cell data is concerning, especially if they are being coerced to the same distribution as the bulk data. It would be very courageous to assume that the true distribution of expression values in each single cell should have the same shape as that of the bulk sample; this assumption will be immediately violated by strong differential expression in a heterogeneous population. Perhaps the authors might consider more bespoke single-cell normalization approaches like scran and scNorm.

Response: We thank the reviewer for bringing up this important technical question. As discussed in our response to Reviewer 1's comment, we tested the performance of our method using various normalization methods, including Scran, Seurat, and SCnorm using both simulated and real data. For simulated data, we found that the performance is best when using SCnorm to normalize the data. For real data, DWLS is robust with respect to the choice of normalization methods and is the only method that can recapitulate the correct trend due to treatment.

Reviewers' comments:

Reviewer #1 (Remarks to the Author):

The authors performed additional experiments using simulation data based on a published scRNA-seq simulator (Splatter). The new results support the robustness of their method (and indeed other methods) against dropout rates. DWLS is indeed slightly more accurate across different dropout rate.

In terms of robustness against normalisation methods, the new results based on Splatter simulation data suggest that the cell-type proportion estimation can be greatly affected by the use of different normalisation methods. If I am reading Fig S3a-c correctly, the results of the estimation of the proportion of minor cell-type (those that simulated to be 5%) can be grossly over-estimated (estimated to be 25-50% by DWLS using Seurat and Scran normalised data; Fig S3a-b). The estimation of the proportion of the major cell type (those comprising 50% of the cells) performs consistently well across all methods and normalisation methods (Fig S3c). Considering that Seurat is arguably the most popular pipeline for scRNA-seq, their results does NOT support the claim that DWLS can accurately estimate the proportion of rare cell-type (defined as those with 5%). The authors did not discuss this result in the main text, instead simply focusing on the fact that SCNorm is better than the other normalisation method, which was not the point of their paper.

Minor comment: The formatting of the in-text citation clearly needs to be updated

Reviewer #2 (Remarks to the Author):

The authors have addressed the concerns I raised previously. However, Reviewer 1 raises some interesting points.

Generally speaking, Reviewer 1 is right to be skeptical. seurat/scran perform a simple scaling normalization while SCnorm performs non-linear normalization, analogous to the difference between TMM normalization and quantile normalization for bulk RNA-seq data. While only SCnorm is capable of removing non-linear biases, it requires stronger assumptions to do so. For example, SCnorm makes the implicit assumption that most genes at any given abundance are not differentially expressed, while the scaling methods only assume that most genes are not DE. To see the difference, you can imagine a situation where many high-abundance genes are strongly DE in a cell population; non-linear methods would potentially remove that difference and discard biological information, whereas scaling methods would be fine.

So, the choice of normalization can matter. Now, the authors claim that their method is doing "better" with SCnorm-normalized data - is this because the normalization is more accurate, which helps downstream processing; or is it because both SCnorm and the proposed method are wrong in ways that happen to cancel each other out, which obviously would not be particularly rigorous? The authors do not distinguish between these two possibilities, and Reviewer 1 is correct to hassle them on that point. On a practical note, scaling normalization is by far the most common class of normalization strategies, so poor performance on scale-normalized data is a concern.

It would be prudent for the authors to determine exactly why their method behaves as it does. For example, does SCnorm change the expression of a few high-abundance genes to drive a change in the estimated proportions? And if SCnorm is performing an inappropriate change, does it make sense to be supplying incorrect values to the deconvolution method, even if it "improves" the accuracy of the

downstream estimates? For example, I would start by taking the simulated counts and obtaining the "true" normalized values (e.g., by dividing by the true scaling factor for each cell), so as to determine the effect of supplying perfectly normalized data.

Point by point response to reviewers' comments.

We thank the reviewers for their thorough reading of our manuscript and constructive suggestions. Our point by point response to their comments is described below.

Reviewer #1 (Remarks to the Author):

The authors performed additional experiments using simulation data based on a published scRNA-seq simulator (Splatter). The new results support the robustness of their method (and indeed other methods) against dropout rates. DWLS is indeed slightly more accurate across different dropout rate.

Response: We thank the reviewer for bringing up this important issue in his/her previous review which has helped us improve the quality of this manuscript.

In terms of robustness against normalisation methods, the new results based on Splatter simulation data suggest that the cell-type proportion estimation can be greatly affected by the use of different normalisation methods. If I am reading Fig S3a-c correctly, the results of the estimation of the proportion of minor cell-type (those that simulated to be 5%) can be grossly over-estimated (estimated to be 25-50% by DWLS using Seurat and Scran normalised data; Fig S3a-b). The estimation of the proportion of the major cell type (those comprising 50% of the cells) performs consistently well across all methods and normalisation methods (Fig S3c). Considering that Seurat is arguably the most popular pipeline for scRNAseq, their results does NOT support the claim that DWLS can accurately estimate the proportion of rare cell-type (defined as those with 5%). The authors did not discuss this result in the main text, instead simply focusing on the fact that SCNorm is better than the other normalisation method, which was not the point of their paper.

Response: We agree with the reviewer's interpretation that different normalization methods can result in variation of deconvolution accuracy. Among the three methods tested here, SCNorm seems to perform better than the other methods. To understand why this is the case, we examined the mathematical details of each method and noticed a crucial difference: while SCNorm directly normalizes sequence read counts, both Seurat and Scran log-transform after normalization. To test if the reason for the poorer performance was caused by log-transformation, we reanalyzed the simulated and ISC data by applying Seurat and Scran without log-transformation. We found that the performance is indeed much improved (Supplemental Figure 3 a,b,c). As such, we think our method is robust to normalization methods if the data are scaled properly. We have added this discussion in our revised manuscript.

Minor comment: The formatting of the in-text citation clearly needs to be updated

Response: We have fixed the format of the in-text citation as suggested, to match standard *Nature* citation formatting.

Reviewer #2 (Remarks to the Author):

The authors have addressed the concerns I raised previously.

However, Reviewer 1 raises some interesting points. Generally speaking, Reviewer 1 is right to be skeptical. *seurat*/*scran* perform a simple scaling normalization while *SCnorm* performs non-linear normalization, analogous to the difference between TMM normalization and quantile normalization for bulk RNA-seq data. While only *SCnorm* is capable of removing non-linear biases, it requires stronger assumptions to do so. For example, *SCnorm* makes the implicit assumption that most genes at any given abundance are not differentially expressed, while the scaling methods only assume that most genes are not DE. To see the difference, you can imagine a situation where many high-abundance genes are strongly DE in a cell population; non-linear methods would potentially remove that difference and discard biological information, whereas scaling methods would be fine.

So, the choice of normalization can matter. Now, the authors claim that their method is doing "better" with *SCnorm*-normalized data - is this because the normalization is more accurate, which helps downstream processing; or is it because both *SCnorm* and the proposed method are wrong in ways that happen to cancel each other out, which obviously would not be particularly rigorous? The authors do not distinguish between these two possibilities, and Reviewer 1 is correct to hassle them on that point. On a practical note, scaling normalization is by far the most common class of normalization strategies, so poor performance on scale-normalized data is a concern. It would be prudent for the authors to determine exactly why their method behaves as it does. For example, does *SCnorm* change the expression of a few high-abundance genes to drive a change in the estimated proportions? And if *SCnorm* is performing an inappropriate change, does it make sense to be supplying incorrect values to the deconvolution method, even if it "improves" the accuracy of the downstream estimates? For example, I would start by taking the simulated counts and obtaining the "true" normalized values (e.g., by dividing by the true scaling factor for each cell), so as to determine the effect of supplying perfectly normalized data.

Response: We thank the reviewer for his/her comment and suggestion. As discussed in our response to Reviewer #1's comments, we examined the mathematical details for each normalization method and found that, while *SCnorm* directly normalizes UMI counts, both *Seurat* and *Scran* first normalize the data and then log-transform them. Because the distribution is distorted by log-transformation, the deconvolution accuracy is reduced. By re-analyzing the simulated and ISC data without log-transform, we

showed that our method is robust against normalization methods provided the data are scaled properly (Supplemental Figure 3g).

REVIEWERS' COMMENTS:

Reviewer #1 (Remarks to the Author):

The authors have addressed my concerns.